# The Changing Detection Rate of Respiratory Syncytial Virus in Adults in Western Australia between 2017 and 2023

**DOI:** 10.3390/v16050656

**Published:** 2024-04-23

**Authors:** David A. Foley, Cara A. Minney-Smith, Andrew Tjea, Mark P. Nicol, Avram Levy, Hannah C. Moore, Christopher C. Blyth

**Affiliations:** 1Department of Microbiology, PathWest Laboratory Medicine WA, Nedlands, WA 6009, Australiaandrew.tjea@health.wa.gov.au (A.T.); christopher.blyth@uwa.edu.au (C.C.B.); 2Wesfarmers Centre of Vaccines and Infectious Diseases, Telethon Kids Institute, University of Western Australia, Perth, WA 6009, Australia; mark.nicol@uwa.edu.au (M.P.N.); hannah.moore@telethonkids.org.au (H.C.M.); 3School of Medicine, University of Western Australia, Perth, WA 6009, Australia; 4Marshall Centre, Biomedical Sciences, University of Western Australia, Perth, WA 6009, Australia; 5School of Population Health, Curtin University, Perth, WA 6009, Australia; 6Department of Infectious Diseases, Perth Children’s Hospital, Nedlands, WA 6009, Australia

**Keywords:** respiratory infection, respiratory syncytial virus, non-pharmaceutical intervention, SARS-CoV-2, respiratory virus infection, adults, seasonality

## Abstract

The incidence of respiratory syncytial virus (RSV) in adults is inadequately defined and the impact of SARS-CoV-2-related non-pharmaceutical interventions (NPIs) is underexplored. Using laboratory data, we described the detection rate of RSV in adults ≥16 years in Western Australia (WA) between 2017 and 2023. With the exception of 2020, RSV detections rose annually between 2017 and 2023, reaching 50.7 per 100,000 in 2023 (95% confidence interval [CI], 47.9–53.8). RSV testing expanded considerably across the study period, with the testing in 2023 more than five times the 2017 total. The detection rate was highest in adults ≥60 years between 2017 and 2019, particularly those ≥75 years. Following 2020, the detections in all age groups increased, with the highest detection rate in 2023 in those ≥75-years (199.5 per 100,000; 95% CI, 180.5–220). NPIs significantly impacted RSV seasonality; the preceding winter pattern was disrupted, resulting in an absent 2020 winter season and two major summer seasons in 2020/21 and 2021/22. The RSV season began to realign in 2022, reverting to a winter seasonal pattern in 2023 and the largest season in the study period. Ongoing surveillance will be required to understand the stability of these increases and to delineate the impact of new immunisation strategies.

## 1. Introduction

Respiratory syncytial virus (RSV) is a common cause of acute respiratory tract infection in adults, ranging from mild upper respiratory symptoms to pneumonia [1]. Exposure and reinfection are common throughout life, with older age and immunosenescence risk factors for more severe disease [2]. Individuals with altered immunity or lower physiological reserves, those residing in higher-risk settings such as residential care facilities, and First Nations people are also at an increased risk of severe disease [2,3]. Global estimates indicate that approximately 1.5 million RSV infections occur annually in individuals aged ≥50 years in industrialised countries, with over 210,000 receiving hospital-level care [2]. In contrast to the low mortality associated with RSV infection in children, the case fatality in adults ≥60 years and those with predisposing conditions is estimated to exceed 8% [1].

Despite the evident significance for RSV in adults, the true burden and impact of infection remain poorly defined, primarily due to case under-ascertainment secondary to low routine testing and the limitations of published clinical and outcome data [2]. The introduction of new RSV immunisations for adults has underlined these gaps [4]. There is a need for comprehensive data to identify groups that may benefit from these interventions and to assess their effectiveness post-implementation. The increased availability and use of testing for RSV, linked to the emergence of severe acute respiratory syndrome coronavirus 2 (SARS-CoV-2) [5], has provided an opportunity to better understand the incidence of RSV infection in adults.

Non-pharmaceutical interventions (NPIs) associated with SARS-CoV-2 altered the epidemiology of RSV and other respiratory viruses [5,6,7,8]. The SARS-CoV-2 pandemic in Western Australia (WA) was marked by stringent state border controls and proactive public health measures. Border closures and other internal NPIs contributed to the successful elimination of SARS-CoV-2 in WA between May 2020 and March 2022. Outside of a stay-at-home order in early 2020, there were minimal internal infection prevention measures in 2020, with no mandated mask requirements or restrictions to large gatherings [9]. Short intervals of increased NPIs were introduced in response to isolated community transmission events of SARS-CoV-2 in 2021 and 2022. There was an absent RSV winter season in WA in 2020, followed by a subsequent inter-seasonal summer surge associated with a collapse in RSV genomic diversity [10]. Similar resurgences in other jurisdictions have been reported, linked to an expanded group of RSV-naïve children [11]. Although these changes have been well described in children, there are limited data characterising the impact of these measures on RSV epidemiology in adults.

This report aims to describe the detection rate of RSV in adults in WA and investigate the impact of SARS-CoV-2 and associated NPIs on RSV seasonality and ascertainment.

## 2. Methods

### 2.1. Location

WA is the largest state in Australia, covering 2.6 million square kilometres. The majority of the 2.1 million adults (≥16 years) reside in metropolitan Perth (1.7 million) and the southern region (357,000), with the remainder residing in the northern region (68,000). WA encompasses a variety of climates, with a temperate climate in metropolitan Perth and the southern region and a tropical climate in the northern region. The temperate areas experience the coolest months from June to August, coinciding with the southern hemisphere winter—conversely, the northern tropical region experiences sub-tropical to tropical climates with a rainy season from November to April [12].

### 2.2. Laboratory Data

RSV detection data between 1 January 2012 and 31 December 2023 and RSV testing data between 1 January 2017 and 31 December 2023 for individuals ≥16 years of age were obtained from PathWest Laboratory Medicine WA, a state public reference laboratory network. The network predominantly provides testing for emergency departments and hospitals, with few referrals from primary care. Molecular RSV testing was performed over this period utilising a combination of both commercial platforms (BioFire respiratory panel [BioMérieux, Marcy-l’Etoile, France], Xpert Flu/RSV assay [Cepheid, Sunnyvale, California, America] and PlexPCR RespiVirus [SpeeDx, New South Wales, Australia]), and an in-house assay [13]. Repeat negatives in the same individual were counted once-weekly, with repeat detections counted once per 3-month period. The additional data obtained included the collection date, age, and the residential postcode at the time of sampling.

### 2.3. Analysis

The postcodes were grouped into three geographic areas, determined by residential distribution and climate [12]: metropolitan Perth, the southern region (the Midwest, Goldfields, Wheatbelt, Southwest, and Great Southern regions), and the northern region (Kimberley and Pilbara). Individuals with an out-of-state postcode or where a postcode was unavailable were excluded from the location-based analysis.

The pre-SARS-CoV-2 RSV average epidemic curve was calculated using the 2012 and 2019 detection data and the World Health Organisation method [14]. A retrospective definition was used to delineate seasonality, with the first two weeks above the detection threshold of 1.2% of the total RSV detections across the calendar year set as the season onset and the final week above this threshold as the offset, with allowance for a single discrepant week [15].

The statistical analysis was conducted using Stata/IC, V.11.2 statistical software (Stata Corp, College Station, TX, USA). The groups were compared using Chi-squared for proportions and the Mann–Whitney U test for non-normally distributed continuous variables. A *p*-value *p* < 0.05 was considered statistically significant. The Australian Bureau of Statistics data were used as the population denominators for the detection rate, where rates were expressed per 100,000 population [16]. The Australian Bureau of Statistics provides age-based population estimates in 3-month intervals. The larger population was used to calculate the detection rate when the RSV season spanned two or more population estimate intervals. The detection rates between 2017 and 2019 were combined as a baseline period for comparison with those between 2020 and 2023, expressed as detection rate ratios (DRRs) with 95% confidence intervals.

## 3. Results

There were 185,416 tests for RSV between 1 January 2017 and 31 December 2023, and RSV was detected in 4743 (2.5% positivity; 95% CI, 2.5 to 2.6).

### 3.1. RSV Seasonality in the Baseline Period between 2017 and 2019

During the baseline period, RSV detections displayed a consistent winter seasonal pattern with a mid winter peak, aligning with the 2012–2019 average epidemic curve (Appendix A). The RSV season in this period typically began in April and lasted until October (Table 1). The 2019 season was shorter but had more tests and detections than 2017 and 2018. Although metropolitan Perth accounted for the largest portion of detections each season, the seasonal detection rate of RSV between 2017 and 2019 was higher in the northern region (38.2 per 100,000; 95% CI, 24.9 to 55.9) and southern region (29.7 per 100,000; 95% CI, 24.3 to 35.9), compared to metropolitan Perth (13.4 per 100,000; 95% CI, 11.7 to 15.3 [Appendix A).

### 3.2. SARS-CoV-2-Related Non-Pharmaceutical Interventions and RSV Seasonality

An increase in RSV detections was associated with a spike in testing in early 2020 (week 9), contemporaneously with the first local SARS-CoV-2 detection, forming a minor RSV season lasting six weeks (Figure 1 and Appendix A). RSV detections decreased from April 2020, associated with the introduction of the stay-at-home order and remained low despite the subsequent sequential lifting of restrictions. Although sporadic RSV detections in mid 2020 were observed, the winter RSV season was absent. In late September (week 39) in 2020, during the Southern Hemisphere spring, RSV detections increased in the southern region, forming a second minor RSV season lasting three weeks.

The primary 2020/21 season began in late October 2020 and continued throughout the Southern Hemisphere summer (Figure 1). A high number of detections were observed in the tropical northern region in early 2021, coinciding with the regional wet season. The end of the 2020/2021 season in mid April 2021 coincided with a more extended period of increased NPIs. The 2020/21 season had the highest percentage positivity in the study period (6.7%; 95% CI, 6.1 to 7.2). The northern region had disproportionately higher seasonal detection rates (Appendix A), compared with the seasonal detection rates in the northern region in the 2017 to 2019 baseline period (DRR = 7; 95% CI, 4.6 to 11 [Appendix A).

The 2021/22 RSV season was again observed during the summer in the Southern Hemisphere (Table 1, Figure 2). The season offset was associated with two periods of increased NPIs, with the associated public health measures predominantly focused on metropolitan Perth and the southern region: the initial phase in response to a SARS-CoV-2 community transmission event and the second extended phase connected to the full re-opening of state borders and the transition to endemic transmission of SARS-CoV-2. There was persistent detection of RSV in the tropical northern region in early 2022 through the latter part of the 2021/22 RSV season (Figure 2 and Appendix A). This sustained detection of RSV coincided with the regional wet season.

The 2022 season began in early August, towards the end of the Southern Hemisphere winter (Table 1, Figure 2). Although this season was the shortest primary season observed, the total RSV detections and mean detections per week were the second highest in the study period. The 2023 season began in late May 2023, marking a reversion to the winter seasonal pattern. This season, spanning 20 weeks, had the highest total and mean detections/week in the study period.

### 3.3. RSV Detection Rates per Year

RSV detection rates per year in adults in WA increased from 16.3 (95% CI, 14.6 to 18.1) in 2017 to 26.7 (95% CI, 24.6 to 29) per 100,000 population in 2019. Following the absent winter season in 2020, the annual detection rate of RSV significantly increased to more than double the 2017 to 2019 baseline detection rate (Appendix A; DRR = 2.4 in 2023; 95% CI, 2.2–2.7). Similar increases were observed in all three regions when compared to the 2017 to 2019 baseline detection rates in the respective region, with the northern region reaching its highest rate in 2021 (DRR = 5.9; 95% CI, 4.2 to 8.7), whilst metropolitan Perth peaked in 2023 (DRR = 3.1; 95% CI, 2.8 to 3.5).

### 3.4. Testing and Percentage Positivity

Similar numbers of RSV tests were performed in 2017 and 2018. Testing notably increased in 2019 (Table 2) and each year thereafter, reaching the highest number in 2023. The annual test percentage positivity was highest between 2017 and 2019, decreasing in 2020 and increasing again in 2021, until declining to a plateau in 2022 and 2023 (Figure 3). A similar pattern was observed in metropolitan Perth and the southern region. The northern region had the highest percentage positivity in 2021 (6.5%; 95% CI, 5.7% to 7.4%) of the study period.

### 3.5. RSV Testing and Detections by Age Group

The ≥16 to ≤39 years age group had the lowest detections and detection rate per year, followed by the ≥40 to ≤59 years age group (Table 2). The test percentage positivity was the lowest in the ≥16 to ≤39 years and ≥40 to ≤59 years age groups, compared to the other age groups, between 2017 and 2019. The RSV detections in these age groups more than doubled between 2021 and 2023 (Appendix A), compared with the 2017 to 2019 period. The test percentage positivity was lowest in 2020, increasing in 2022 and 2023, but remained below the levels observed between 2017 and 2019 (Figure 3), consistent with the expanded testing in these age groups.

The ≥60 to ≤74 years and ≥75 years age groups consistently had the most RSV detections and the highest detection rate observed between 2017 and 2019, particularly in those aged ≥75 years (86.5 per 100,000 population; 95% CI, 72.6 to 102.4 [Appendix A). The test percentage positivity was highest between 2017 and 2019 in the ≥60 to ≤74 years and ≥75 years age groups, compared to the other age groups. The detection rate in the ≥60 to ≤74 years and ≥75 years age groups increased in 2021 and reached the highest rate in the study period in 2023 (Table 2 and Appendix A), more than double the 2017 to 2019 baseline detection rate in these age groups (Appendix A: ≥60 to ≤74 years age group, DRR = 2.3; 95% CI, 1.9 to 2.9; and 75 years age group, DRR 2.3; 95% CI, 1.9 to 2.8). The ≥75 years age group had the highest detection rate in the study period in 2023 at 199.5 per 100,000 population per year (95% CI, 180.5 to 220 [Appendix A). The testing increase in 2022 and 2023 was associated with a drop in the test percentage positivity below the proportions between 2017 and 2019, but remained above the other age groups.

## 4. Discussion

The RSV detection rate rose each year between 2017 and 2019. Following a decrease in 2020, the RSV detection rate continued to rise, reaching 50.7 per 100,000 population in 2023. The RSV testing increased markedly across the study period, contributing to this increase; the testing in 2023 was more than five times higher than the 2017 total. The ≥60 to ≤74 years and ≥75 years age groups had the highest detection rate between 2017 and 2019. Again, following 2020, the detection rate in all age groups increased, with the ≥75 years age group having the highest rate at 199.5 per 100,000 per year in 2023. Although the detections increased in all regions, disproportionately higher rates were observed in the tropical northern region. The RSV seasonality in WA was disrupted by SARS-CoV-2-related NPIs, resulting in an absent winter season in 2020 and two major summer RSV seasons in 2020/21 and 2021/22. RSV seasonality began to realign with the baseline seasonality in 2022, with a reversion towards the winter seasonal pattern in 2023. The 2022 and 2023 RSV seasons had the highest detection number and most detections per week.

With the exception of 2020, the annual RSV detection rate in WA increased yearly to 50.7 per 100,000 adults per year in 2023. Comparable detection data from other regions for this age range are not readily available [17]. These rates are 100 times lower than the internationally reported rates in children under five years [18]. Importantly, these rates encompass a larger population and underscore the significant burden of medically attended RSV in adults in our region. Notably, RSV testing considerably increased over the study period, contributing to this continual rise in detections. This expansion of testing has uncovered the significant under-ascertainment of RSV infection in adults in our region. This increase in testing can be partly attributed to local SARS-CoV-2-related hospital testing to guide hospital infection control decisions and the provision of antiviral therapy, with the state reference laboratory predominately employing a respiratory panel that includes SARS-CoV-2 and RSV for these requests. Although yearly test percentage positivity decreased, in-season percentage positivity remained high, highlighting the within-season burden of infection.

Individuals ≥60 years old had the highest baseline detection rate, especially those ≥75 years. These age groups have been broadly recognised as being at risk of more severe RSV infection [19]. Comparing rates of RSV in adults ≥60 years across settings is difficult; population sociodemographic constructs, environmental conditions, testing strategies, data sources, and analysis approaches influence the rates [19]. However, the RSV hospitalisation attack rate has been reported to be between 1 and 1.5 per 1000 per year [19]. The detection rates between 2017 and 2019 were lower than these numbers, likely linked to under-ascertainment. Furthermore, although PathWest Laboratory Medicine has a broad footprint in WA, these data do not fully represent the population burden due to the absence of private pathology results. The detection rate increased to 77.7 per 100,000 population in the ≥60 to ≤74 years age group and 199.5 per 100,000 population in the ≥75 years in 2023, rates more in-line with the published data [19]. This increase is expected to be driven by improved case ascertainment secondary to expanded respiratory virus testing. However, the decrease in population-level immunity and the associated stronger chains of transmission following the attenuated preceding RSV seasons is expected to be a factor. Close monitoring over the next few years will be critical to understanding the stability of these changes. Notwithstanding, these data highlight the high detection rate of RSV and emphasise the potential value of immunisations in this at-risk population. The steps to provide access to the newly licensed RSV vaccines for older adults have already begun in our region.

There were differences in the detection rate of RSV in WA between regions prior to the emergence of SARS-CoV-2, with a higher detection rate observed in the northern tropical region. A significant portion of the population in this region has pre-existing health conditions and faces sociodemographic inequalities [20]. The northern region also experienced disproportionately higher rates of RSV detections during the 2020/21 and 2021/22 seasons. Although this region had travel restrictions in place, it did not undergo the same extent of increased NPIs as the metropolitan and southern region, which may have facilitated this increase. Further, these summer seasons occurred during the northern tropical region’s wet season, with environmental conditions more favourable to RSV transmission [21]. RSV activity peaks in tropical areas during high humidity, primarily during the rainy season [22]. However, a rainy season RSV peak was not previously observed in northern WA [12] until the disrupted seasons in 2020/21 and 2021/22. This more favourable transmission period may have been unmasked by the lower asynchronous viral migration pressures from the neighbouring, more populous region. The underlying differences in at-risk groups and the potential differences in the timing of peak RSV activity in tropical regions will necessitate consideration of the development of individualised and bespoke RSV immunisation strategies.

These data reiterate the impact of SARS-CoV-2-related NPIs on RSV seasonality. The implementation of NPIs in early 2020 contributed to a marked reduction of RSV detections in WA. Despite minimal internal restrictions, the absence of the winter RSV season in 2020 was likely associated with a preceding reduction in endemic RSV circulation and the maintenance of state border restrictions preventing the introduction of new viral strains. The emergence of a single clade during the 2020/21 summer season, previously described in children [10] and now in adults, has been postulated to be secondary to a combination of minimal NPIs, an expanded RSV-naïve infant cohort [11], and a reduction in population-level immunity fostering stronger chains of transmission. The 2020/21 and 2021/22 RSV seasons were also disrupted by periods of NPIs, potentially moderating their intensity and duration, with the ending of both seasons associated with periods of increased NPIs. These early terminations and unfavourable summer environmental conditions may have led to an increased at-risk population for the subsequent 2022 and 2023 seasons.

The 2022 season, marked by its delayed start, followed the lifting of state border restrictions and a period of heightened NPIs associated with the shift towards SARS-CoV-2 endemicity. The higher detection rates during this season correlated with more favourable environmental conditions and, importantly, was the first season post-2019 uninterrupted by periods of increased NPIs. In 2023, the RSV season fully reverted to its traditional winter timing and had the highest number of detections in the study period. This surge in detections is likely secondary to a confluence of factors: an expanded population at risk from the preceding absent and attenuated seasons, unmitigated viral introduction into the region, and optimal environmental conditions for transmission.

There are several limitations to this report. The absence of private pathology data will contribute to an underestimation of the detection rate of RSV in WA. Differing health-seeking behaviours and associated testing behaviours at baseline and over the study period between the regions and age groups may have contributed to unaccounted differences in the detection rates [20]. The optimal approaches to defining the season onset and offset are uncertain [15]. This method was employed due to the stability across periods where there are shifts in RSV detections, significant temporal changes in testing practices, and marked fluctuations in the prevalence of other respiratory viruses. The approach used to define the season onset and offset was applied across the years, contributing to differing onset and offset thresholds for the summer RSV 2020/21 and 2021/22 seasons. Specifically, the lower total annual detections in 2020 contributed to a lower onset threshold, which contributed to the formation of minor seasons and an earlier onset for the 2020/21 season. Further, a shift in season intensity and peak activity may artificially shorten the season even when activity continues at significant levels.

## 5. Conclusions

A high detection rate of RSV infection was observed in adults in Western Australia, particularly those ≥60 years. Expanded respiratory virus testing has played an essential role in improving our understanding of the detection of RSV in our region. Further research is required to explore the true burden of infection spanning both the community and hospital and across the spectrum of severity, to inform and evaluate RSV prevention strategies.

## Figures and Tables

**Figure 1 viruses-16-00656-f001:**
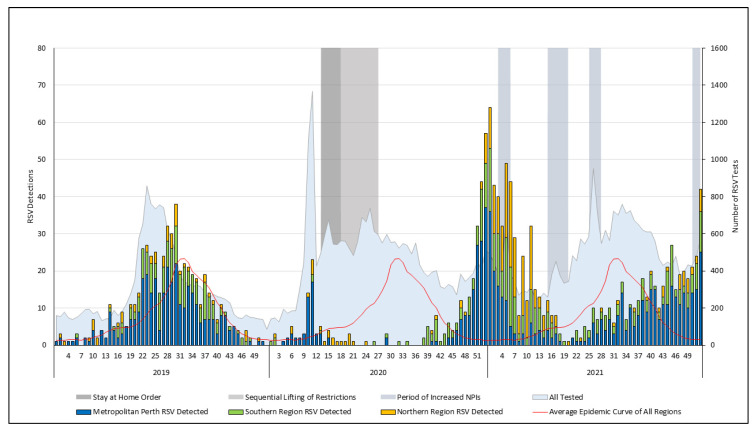
RSV testing and detections per week by region between 2019 and 2021, with the associated SARS-CoV-2-related non-pharmaceutical interventions. The average epidemic curve was calculated using RSV detection data between 2012 and 2019. Overview of non-pharmaceutical interventions: the first local case of SARS-CoV-2 was in week eight of 2020; a state-wide stay-at-home order (weeks 14–17 of 2020) was followed by the sequential lifting of restrictions (18–26 weeks), with gatherings of 10 allowed at week 18, 20 at week 21, and 100 at week 24; there were periods of increased NPIs that included mask requirements in the metropolitan and southern region: in weeks 4 and 5, weeks 16 to 19, and 26 to 27 in 2021; and week 50 in 2021 to week 2 in 2022. RSV, respiratory syncytial virus.

**Figure 2 viruses-16-00656-f002:**
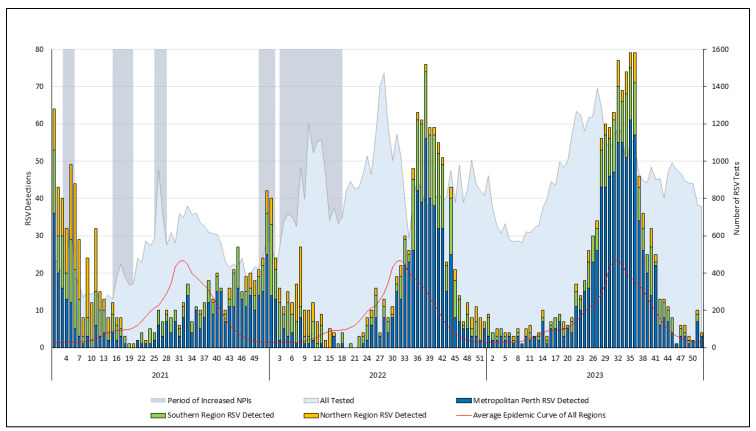
RSV testing and detections per week by region between 2021 and 2023 with the associated periods of SARS-CoV-2-related non-pharmaceutical interventions. The average epidemic curve was calculated using RSV detection data between 2012 and 2019. Overview of non-pharmaceutical interventions: there were periods of increased NPIs that included mask requirements in the metropolitan and southern region from week 50 in 2021 to week 2 and week 3 to 17 in 2022; border restrictions were fully removed in week 9 of 2022; and there was sporadic community detection of SARS-CoV-2 during the periods of increased NPIs in 2022. Continued low-level local transmission of SARS-CoV-2 was observed in January 2022, exceeding one hundred local cases per day by week 6 of 2022. RSV, respiratory syncytial virus.

**Figure 3 viruses-16-00656-f003:**
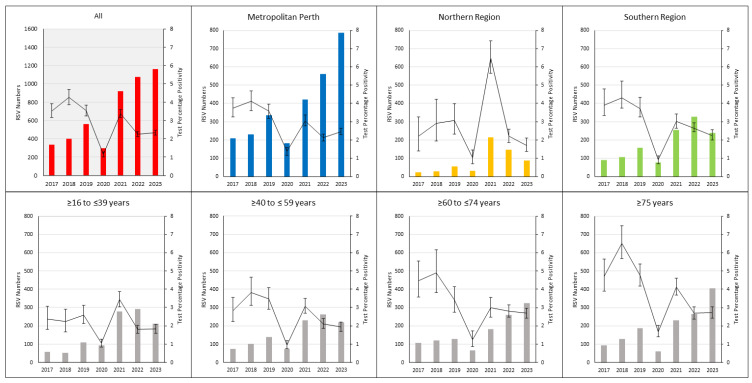
RSV detections and test percentage positivity, with a 95% confidence interval, by region and age group between 2017 and 2023. RSV, respiratory syncytial virus.

**Table 1 viruses-16-00656-t001:** Comparison of the duration, detections, and total tests of RSV seasons between 1 January 2017 and 31 December 2023.

Seasons	2017	2018	2019	2020/21	2021/22	2022	2023
Timing	Week 19 to 44	Week 12 to 42	Week 17 to 42	Week 44 to 15	Week 32 to 8	Week 31 to 46	Week 22 to 43
Duration (weeks)	32	31	26	25	29	16	22
Onset Date	8 May 2017	19 Mar 2018	22 April 2019	26 October 2020	9 August 2021	1 August 2022	29 May 2023
Offset Date	5 November 2017	21 October 2018	20 October 2019	18 April 2021	27 February 2022	20 November 2022	29 October 2023
Detections; n	284	344	482	642	542	688	949
Mean Detections/Week	9	11	19	26	19	43	44
Total Tests; n	6424	6415	11,635	9654	17,687	14,221	23,146
% Positive (95% CI)	4.4 (3.9 to 5)	5.4 (4.8 to 6)	4.1 (3.8 to 4.5)	6.7 (6.1 to 7.2)	3.1 (2.8 to 3.3)	4.8 (4.5 to 5.2)	4.1 (3.8 to 4.4)

95% CI, 95% confidence interval; and RSV, respiratory syncytial virus.

**Table 2 viruses-16-00656-t002:** RSV testing, detections, and detection rate per 100,000 per year between January 2017 and 31 December 2023.

Year	2017	2018	2019	2020 *	2021 *	2022 *	2023
Detections; n	333	400	562	294	919	1077	1158
Total Tests; n	9475	9403	15,910	25,643	27,228	47,814	49,943
% Positive (95% CI)	3.5 (3.1 to 3.9)	4.3 (3.8 to 4.7)	3.5 (3.2 to 3.8)	1.1 (1 to 1.3)	3.4 (3.2 to 3.6)	2.3 (2.1 to 2.4)	2.3 (2.2 to 2.5)
	Detection rate per year (95% CI)	Detection rate per year (95% CI)	Detection rate per year (95% CI)	Detection rate per year (95% CI)	Detection rate per year (95% CI)	Detection rate per year (95% CI)	Detection rate per year (95% CI)
All	16.3 (14.6 to 18.1)	19.3 (17.5 to 21.3)	26.7 (24.6 to 29)	13.7 (12.2 to 15.3)	42.2 (39.5 to 45)	48.7 (45.9 to 51.7)	50.7 (47.9 to 53.8)
By Region							
Metropolitan Perth	12.4 (10.7 to 14.2)	13.7 (12 to 15.5)	19.8 (17.8 to 22.1)	10.8 (9.3 to 12.5)	25 (22.6 to 27.5)	33.2 (30.6 to 36.1)	46.6 (43.4 to 49.9)
Northern Region	35.2 (22.6 to 52.4)	41.1 (27.3 to 59.4)	82.2 (62.1 to 106.8)	45.5 (30.9 to 64.6)	314.2 (273.5 to 359.2)	215.8 (182.3 to 253.6)	127.7 (102.3 to 157.5)
Southern Region	25.8 (20.8 to 31.6)	29.7 (24.3 to 35.9)	44.3 (37.6 to 51.7)	21.3 (16.8 to 26.7)	71.7 (63.2 to 81.1)	91.9 (82.2 to 102.4)	66.7 (58.5 to 75.7)
By Age Group							
≥16 to ≤39 years	6.6 (5.0 to 8.5)	6 (4.5 to 7.9)	12.2 (10.0 to 14.7)	10.4 (8.4 to 12.7)	31.2 (27.6 to 35.1)	32.5 (28.9 to 36.5)	22.6 (19.6 to 25.8)
≥40 to ≤59 years	11.2 (8.8 to 14)	14.9 (12.1 to 18.1)	20.3 (17.1 to 24)	10.6 (8.3 to 13.3)	32.6 (28.5 to 37.1)	36.6 (32.3 to 41.4)	30.1 (26.2 to 34.3)
≥60 to ≤74 years	31.2 (25.5 to 37.7)	33.4 (27.7 to 40.0)	35 (29.2 to 41.5)	16.8 (13.0 to 21.4)	45.2 (38.9 to 52.3)	63.7 (56.2 to 71.9)	77.7 (69.5 to 86.7)
≥75 years	62.2 (50.2 to 76.2)	81.6 (68 to 97.1)	114.2 (98.3 to 131.8)	36.1 (27.7 to 46.3)	127.5 (111.6 to 145.1)	138 (121.8 to 155.7)	199.5 (180.5 to 220.0)

*, RSV season disrupted during this period; 95% CI, 95% confidence interval; and RSV, respiratory syncytial virus.

## Data Availability

Data are available upon reasonable request with the required ethics and governance approvals in place.

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
