# Peer review of "The Changing Detection Rate of Respiratory Syncytial Virus in Adults in Western Australia between 2017 and 2023"

_viruses, 2024, doi:10.3390/v16050656_

Round 1

Reviewer 1 Report

Comments and Suggestions for Authors

Author Response

Response to reviewer one

Thank you for sharing your time and expertise. We appreciate your positive feedback.

  • Impact of NPIs of dynamics of within household transmission (e.g., curfew)

Within household transmission dynamics will be influenced by numerous factors that will vary between countries that are not captured in this study: the number of individuals, their exposure profile outside of the house, the relationship of individuals and how they interact in the household, the size of the physical space and airflow, as well as temperature and humidity. A curfew may marginally increase the risk of within household infection in the short term and impact a smaller number of individuals. It is probable that preventing mixing within the community setting and the introduction of facemasks would create a greater cumulative reduction of RSV transmission. 

However, the level of the data in this report is unable to provide the required granularity to answer the reviewer's question. We are seeking funding to prospectively explore within household transmission of respiratory viruses, to answer these questions.    

  • Increased attention to SARS-CoV-2 driving increased RSV detection in older adults.

We agree that the emergence of SARS-CoV-2 has led to increased detections of RSV. Testing notably increased across the study period. Within RSV season test percentage positivity remained high. This finding suggests that there was an increase in RSV infections in older adults. However, the study construct is not able to fully indicate the relative proportions of this increase (case ascertainment vs true increase). We are in the process of completing a complimentary study using a cohort-based approach to explore these changes further.

Reviewer 2 Report

Comments and Suggestions for Authors

The deployment of non-pharmacological interventions (NPIs) at scale during the COVID-19 pandemic provided a wonderful opportunity to examine the impact of these interventions on other infections. As such, publication of these laboratory data on RSV infection is a useful addition to existing knowledge. Furthermore, the publication of data from a variety of geographic locations and intervention stringencies is helpful for comparison purposes.

Specific comments:

(1) It would be helpful to have more detail on the context, e.g. more detail about the nature and timing of NPIs, in order to better understand the findings. Can the authors speculate on the changes in testing behaviour over the study period? I expect COVID-19 in itself changed testing patterns for all respiratory viruses. 

(2) Is there any chance data from community laboratories can be included? The current data will be very skewed towards more severe disease. It would be a much stronger paper if data were obtained across the whole country, knowing there was state-to-state variation in NPI use.

(3) The paper could be shortened without loos of key information.

Author Response

Response to reviewer two

Thank you for sharing your time and expertise. We appreciate your positive feedback.

  • Context of NPIs (non-pharmaceutical interventions)

We included details of the timing and nature of NPIs that may have been overlooked within the description of the included figures (Figures 1 and 2) and Figure S1 to provide context. Additional information can be added if further details are requested.  

  • Changes in testing behavior over the study period

We agree that testing behaviors changed over the study period. RSV testing markedly increased across the study period. We discuss the drivers of testing within lines 251 to 259 in the discussion and highlighted the unexplored drivers of testing in lines 327 to 330 when exploring the limitations of this dataset.

  • Community data

The inclusion of data from laboratories that have a broader community level provider footprint would greatly strengthen this report. Unfortunately, the prolonged negotiations to obtain access to these data has not been fruitful.  

  • The paper could be shortened without loss of key information.

We reviewed the paper and identified areas to reduce the word count.

Round 2

Reviewer 1 Report

Comments and Suggestions for Authors

I accept author`s reply and I would like to recommend to publish it in the Viruses.